# OpenReview forum: "UpSafe℃: Upcycling for Controllable Safety in Large Language Models"
_ICLR.cc/2026/Conference — Submitted to ICLR 2026_

### Official Review · Reviewer_sSyc · 2025-10-17

**Soundness:** 3
**Presentation:** 3
**Contribution:** 3
**Rating:** 6
**Confidence:** 4

**Summary:**

The paper introduces UPSAFE°C, a method that improves the safety of pretrained LLMs by upcycling harmful latent concepts into benign ones using residual stream alignment. The approach is efficient, interpretable, and shows strong refusal performance across various base models and refusal datasets.

**Strengths:**

1. **Strong Empirical Results.** It shows strong performance across both standard Large Language Models (LLMs) and Large Reasoning Models (LRMs).
2. **Practicality.** The method is practical—it requires minimal model modification and preserves both safety and utility.
3. **Detailed Analysis.** The paper provides thorough ablations and layer-wise visualizations, offering clear insights into how and where safety improvements occur.

**Weaknesses:**

1. **Limited Scope of Evaluated Attacks.** The paper evaluates UPSAFE°C against WildJailbreak, JBB, and StrongReject, but does not test against sophisticated jailbreak methods such as PAIR [1] or ReNeLLM [2]. Evaluation on these stronger attack vectors would better assess robustness.
2. **Weak Baseline Comparisons.** The baselines are limited to vanilla models, SFT-only training, and a single-stage MoE variant. Comparisons against recent defense methods, particularly representation-level [3,4] and reasoning-based [5] safety techniques, would better position the method's effectiveness.

If my concerns are addressed, I’m willing to raise my score.

[1] Chao, Patrick, et al. "Jailbreaking black box large language models in twenty queries." 2025 IEEE Conference on Secure and Trustworthy Machine Learning (SaTML). IEEE, 2025.\
[2] Ding, Peng, et al. "A wolf in sheep's clothing: Generalized nested jailbreak prompts can fool large language models easily."NAACL (2023).\
[3] Zou, Andy, et al. "Improving alignment and robustness with circuit breakers." NeurIPS (2024).\
[4] Yousefpour, Ashkan, et al. "Representation bending for large language model safety." ACL (2025).\
[5] Jeung, Wonje, et al. "SAFEPATH: Preventing Harmful Reasoning in Chain-of-Thought via Early Alignment." NeurIPS (2025).

**Questions:**

1. The paper identifies safety-critical layers empirically through linear probing but does not explain whether there is a consistent pattern for why safety signals concentrate in specific layers.
2. Some utility metrics increase after applying UPSAFE°C. Could the authors explain why performance improves in these cases? (Table 1)
3. According to Figure 4, accuracy drops significantly as τ increases, while Table 1 shows strong utility performance at τ=0.5. Also, where is the vanila model in Figure 4?

---

> ### Author Response · Authors · 2025-11-23
> **Response to Reviewer sSyc**
>
> > **W1 & W2: Limited scope of evaluated attacks and weak baseline comparisons**
>
> We thank the reviewer for these valuable suggestions. For both points, We kindly refer the reviewer to our GR1 for the complete experimental updates and extended analyses.
>
> > **Q1: Explain whether there is a consistent pattern and why safety signals concentrate in specific layers.**
>
> We appreciate the reviewer’s question. Empirically, our experiments show that safety-critical layers are a relatively stable phenomenon across different datasets, the layers exhibiting the strongest harmful-vs-benign separability remain largely consistent (see Fig. 9 in Appendix B.1 for details).
>
> This observation aligns with results from the representation-learning papers [1, 2], which shows that different transformer layers encode different semantic functions. Prior work [3] on safety representations further demonstrates that safety-aligned LLMs tend to form specialized layers where harmful content becomes linearly separable from benign content. However, accurately identifying such layers typically requires heavy analysis costs, and several studies [4, 5] also point out that even known safety layers can be sensitive to latent-space perturbations, making robust identification non-trivial.
>
> In contrast, our approach uses lightweight linear probing to select a suitable safety-critical layer: this requires only a single forward pass to collect intermediate representations and train a set of linear classifiers that serve directly as the initial soft guardrail. During Stage 2, we further strengthen this soft guardrail through joint training with the duplicated safety experts, enhancing its discrimination ability and robustness.
>
> > **Q2: Some utility metrics increase after applying UPSAFE°C. Could the authors explain why performance improves in these cases?**
>
> We appreciate the reviewer’s question. The increase in certain utility metrics can be attributed to the added diversity in model computation paths introduced by the safety experts. Since the original experts are frozen and the safety experts are initialized from them, the model’s general capabilities are preserved. At the same time, routing some inputs through the newly added safety experts creates alternative processing paths, which can sometimes lead to better general task performance—essentially giving the model more flexible ways to handle inputs without compromising safety.
>
> This effect is more consistently observed in reasoning models. The safety experts, initialized from the original experts, capture additional CoT patterns during their training on safety data. These alternative pathways can indirectly improve performance on certain general tasks.
>
> > **Q3: According to Figure 4, accuracy drops significantly as τ increases, while Table 1 shows strong utility performance at τ=0.5. Also, where is the vanila model in Figure 4?**
>
> In Figure 4, the utility is measured by the over-refusal rate on XSTest, as explained in the main text. XSTest contains prompts near the **safety boundary**, making it more sensitive to changes in safety temperature. **Increasing $\tau$ effectively shifts the model's safety boundary, which has a relatively larger impact on these challenging boundary prompts.**
>
> In Table 3 of Appendix B.4, we also show results on more general benchmarks across varying $\tau$. These tasks do not lie near the safety boundary, and therefore their performance remains stable across a broad range of $\tau$, explaining why performance is still strong at $\tau=0.5$.
>
> Regarding the vanilla model in Figure 4, different colors indicate different safety temperature settings, and the point with τ=0 (**black**) corresponds to the vanilla model. We will make this clearer in the revised manuscript.
>
> **Reference:**
>
> [1] Pan, Rui, et al. "Lisa: Layerwise importance sampling for memory-efficient large language model fine-tuning." NeurIPS (2024).
>
> [2] Skean, Oscar, et al. "Layer by layer: Uncovering hidden representations in language models." ICML (2025).
>
> [3] Li, Shen, et al. "Safety layers in aligned large language models: The key to llm security." ICLR (2025).
>
> [4] Li, Tianlong, et al. "Revisiting jailbreaking for large language models: A representation engineering perspective." COLING (2025).
>
> [5] Lin, Yuping, et al. "Towards understanding jailbreak attacks in llms: A representation space analysis." EMNLP (2024).

---

> > ### Comment · Reviewer_sSyc · 2025-11-23
> > **Thank you for rebuttal**
> >
> > Thank you for the rebuttal. Most of my concerns are resolved, especially with the inclusion of strong additional baselines. Please include them in the final version. I raised my score accordingly.

---

> > > ### Author Response · Authors · 2025-11-24
> > >
> > > Thank you very much for your helpful and encouraging review! We will incorporate all the suggested changes into the final version of our paper.

---

### Official Review · Reviewer_TAMx · 2025-10-17

**Soundness:** 2
**Presentation:** 3
**Contribution:** 2
**Rating:** 4
**Confidence:** 4

**Summary:**

This paper introduces UPSAFE°C, a unified framework designed to enhance the controllable safety of Large Language Models (LLMs) through safety-aware upcycling, addressing the limitations of existing safety techniques in balancing safety, utility, and controllability. The framework first identifies safety-critical layers in pre-trained LLMs via a Safety Sensitivity Score, and upcycles these layers into a sparse Mixture-of-Experts (MoE) structure. In this MoE setup, a router acts as a "soft guardrail" to selectively activate the original Multi-Layer Perceptrons (MLPs, serving as general experts) and additional safety experts copied from the original MLPs. To strengthen safety discrimination while preserving general capabilities, a two-stage Supervised Fine-Tuning (SFT) strategy is employed: the first stage trains safety experts and the router on harmful data, and the second stage trains only the router on a mixed dataset of harmful and benign data to refine its input discrimination. A safety temperature mechanism is further introduced for inference-time control, dynamically adjusting the safety-utility trade-off by biasing router logits and scaling probabilities.
Experiments across multiple benchmarks, model types, and scales demonstrate that UPSAFE°C achieves robust safety improvements while maintaining competitive general performance. Ablation studies confirm the efficacy of the proposed mechanisms. The safety temperature mechanism enables fine-grained control, allowing the model to approximate the Pareto-optimal frontier between safety and utility, marking a shift from static alignment to dynamic, modular, inference-aware LLM safety control.

**Strengths:**

- Introduces a targeted safety-critical layer scanning mechanism using the SS-Score, which efficiently identifies layers most responsive to safety signals—this avoids unnecessary parameter tuning across all layers and lays a precise foundation for subsequent upcycling.
- The framework’s two-stage SFT strategy effectively specializes safety experts in mitigating harmful content while enabling the router to discriminate between benign and harmful inputs, striking a robust balance between safety enhancement and general capability preservation.
- The novel safety temperature mechanism provides fine-grained, dynamic control over the safety-utility trade-off at inference time, allowing the model to approximate the Pareto-optimal frontier and adapt to diverse safety requirements without retraining.
- Experiments demonstrate strong generalization across model types, scales, and challenging benchmarks, outperforming baselines in both safety improvements.

**Weaknesses:**

- The evaluation part lacks comparative evaluations against established fine-tuning-based safety enhancement baselines or simple defense techniques like filtering by judgers, which limits the verification of its effectiveness in outperforming widely adopted safety techniques.
- As a practically oriented framework, the manuscript does not provide analyses of application costs (e.g., training/inference time overhead) or assess the additional adaptation efforts required to integrate its MoE-upcycled structure and safety temperature mechanism into modern inference frameworks, leaving uncertainty about its efficiency advantage over simpler safety methods.
- The experimental results include numerous perfect scores (100.00), which are uncommon for models like the Qwen series on standard safety datasets—especially given potential dataset biases. Without detailed explanations for these unusually stable high values (e.g., dataset distribution characteristics or evaluation metric artifacts), the reliability of such results remains unsubstantiated.

**Questions:**

For questions, please kindly refer to the previously identified weaknesses, which highlight several issues that require further resolution or explanation. As this paper presents a technically solid framework, addressing these unresolved issues would help enhance the study’s rigor and persuasiveness and may potentially contribute to a higher score.

---

> ### Author Response · Authors · 2025-11-23
> **Response to Reviewer TAMx**
>
> > **W1: The evaluation part lacks comparative evaluations against established fine-tuning-based safety enhancement baselines or simple defense techniques like filtering by judgers.**
>
> We thank the reviewer for raising this important point. As noted in our GR1, we have added extensive comparisons against more fine-tuning–based safety enhancement baselines, covering both LLM and LRM settings. Here, we further provide a direct comparison with filtering-based defense, i.e., using an external judger.
>
> Specifically, we employ LlamaGuard3-8B to filter inputs on three harmful benchmarks. Inputs judged as unsafe are directly rejected, and only the remaining inputs are passed to the model (Llama-3.1-8b-Instruct) for safety evaluation. The results are shown below:
>
> |      Methods       |   JBB   | StrongREJECT | WildJailbreak |
> |:----------------:|:-------:|:------------:|:--------------:|
> | Filtering-based  |  100    |     100      |     79.60      |
> | Ours (0.5)       |  100    |     100      |     90.80      |
>
> Our method achieves stronger robustness on WildJailbreak. Moreover, unlike LlamaGuard3—which provides an **external**, pre-processing–only safety filter—our approach enhances the model’s **intrinsic** safety. We will add this discussion in the revised manuscript.
>
> > **W2: Provide analyses of application costs and assess the additional adaptation efforts required to integrate into modern inference frameworks**
>
> We appreciate the reviewer’s concern regarding deployment efficiency. In Table 2 of Appendix A.1, we have reported the additional training parameters introduced by upcycling. Here we further provide training/inference time overhead measurements on LLama-3.1-8B-Instruct.
>
> |      Method      |       Training Time          | Inference Throughput |
> |----------------|:-------------------------:|:---------------------:|
> | SFT              | 24.28 min                    | 506 tokens/s          |
> | Ours             | Stage 1: 16.63 min; Stage 2: 12.40 min | 503 tokens/s  |
>
>
> *Inference throughput is measured on an A100 GPU using vLLM over 250 WildJailbreak prompts.*
>
> The results indicate that UpSafe℃ introduces only a small amount of additional training time and only minor degradation in end-to-end throughput under vLLM, confirming that the framework remains efficient and practical for deployment.
>
> **For integration**, we already adapted the UpSafe℃ implementation into the PEFT library and implemented inference compatibility with vLLM (see our **anonymous repository** provided in the REPRODUCIBILITY STATEMENT).  we also plan to extend support to additional open-source inference frameworks.
>
> > **W3: Without detailed explanations for these unusually stable high values**
>
> We appreciate the reviewer's observation. The near-perfect scores on JBB and StrongReject primarily stem from the characteristics of these benchmarks. Both datasets consist of **highly explicit and strongly harmful prompts**, making them relatively easy for aligned LLMs to detect and refuse. In fact, even the vanilla base models already achieve very high scores on these two benchmarks. Moreover, their **limited size** (100 and 313 samples, respectively) further contributes to the stability of the results.
>
> For more rigorously evaluation, we therefore rely on more challenging datasets such as WildJailbreak, where harmful intent is more subtle and diverse.

---

> > ### Comment · Reviewer_TAMx · 2025-11-24
> >
> > Thank you for your thoughtful response to my comments. Major concerns I raised have been adequately addressed. Based on your compelling rebuttal, I am pleased to revise my score upward.

---

> > > ### Author Response · Authors · 2025-11-24
> > >
> > > Thank you for your thoughtful and constructive feedback! We will carefully refine the manuscript according to your suggestions.

---

### Official Review · Reviewer_NXtt · 2025-10-19

**Soundness:** 3
**Presentation:** 3
**Contribution:** 3
**Rating:** 4
**Confidence:** 4

**Summary:**

This paper proposes UpSafe, a framework for upcycling large language models (LLMs) to ensure consistent safety alignment across multiple instruction-tuning or post-finetuning stages. The authors observe that safety degradation frequently occurs when models undergo continued training (e.g., domain adaptation, capability enhancement), because existing safety fine-tuning methods (e.g., DPO, SFT) operate in isolation and fail to preserve alignment consistency across evolving model generations.

**Strengths:**

1. The paper introduces the concept of safety upcycling, which is a creative and underexplored idea—treating alignment as a cumulative inheritance process rather than one-time optimization.

2. The paper is clearly written and well-structured.

3. The proposed method addresses a critical real-world challenge: the loss of safety alignment during continual or multi-stage training—a widely recognized issue in the deployment of foundation models.

**Weaknesses:**

1. Limited novelty beyond structural adaptation. The core idea—analyzing or enforcing safety alignment at the safety layer—has appeared in prior works. This paper primarily transfers that analysis into a Mixture-of-Experts (MoE) architecture, rather than introducing a fundamentally new alignment principle.

2. Potential trade-off in temperature-based defense for complex reasoning tasks. The proposed temperature-scaling defense is effective in simple safety-sensitive or low-reasoning contexts. Still, it could significantly degrade performance on hard reasoning or mathematical problems, where token-level uncertainty is naturally higher.

**Questions:**

See Weaknesses.

---

> ### Author Response · Authors · 2025-11-23
> **Response to Reviewer NXtt**
>
> > **W1: Limited novelty beyond structural adaptation.**
>
> We thank the reviewer for raising this point. While there are studies that identify safety-critical layers, existing methods typically rely on heavy analyses of representations and offer no direct mechanism for turning the identified layers into deployable safety modules.
>
> Our framework introduces a **actionable, and conceptually coherent alignment mechanism**, consisting of:
> 1. **Lightweight safety-layer identification.**
> Instead of complex representational analyses, we adopt linear probing, which requires only a single forward pass to collect layer-wise activations. This makes safety-layer discovery significantly cheaper and more scalable.
>
> 2. **New conceptual role of probing: inspiration for an internal soft guardrail.**
> Instead of using probing solely as an analysis tool, we show that linear probing reveals where harmful and benign representations become most separable. This insight motivates and guides the design of our internal soft guardrail (i.e., the routing mechanism). Thus, probing serves as a principled and lightweight signal for where and how such a guardrail should be instantiated.
>
> 3. **Safety upcycling built directly on top of the soft guardrail.**
> The safety-discriminative signal captured by the soft guardrail directly motivates our safety upcycling design: we separate the original experts (for general capabilities) from newly duplicated safety experts (for safety capabilities). Also, safety temperature emerge naturally from this design, enabling controllable safety rather than static constraints.
>
> This end-to-end chain—lightweight probing → soft guardrail → upcycled safety experts → controllable safety—forms a coherent and novel alignment framework, which transforms representational insights into a practical, modular, and controllable safety system.
>
> > **W2: Potential trade-off in temperature-based defense for complex reasoning tasks.**
>
> We thank the reviewer for highlighting this concern. In **Table 3 of Appendix B.4**, we have evaluated broader utility benchmarks (e.g., math and code generation) under varying safety temperature settings. Empirically, we observe that these capabilities remain stable across temperatures. This is largely because that the original experts—responsible for general reasoning—are kept frozen, and the safety experts are initialized from the same weights. As a result, the model’s inherent reasoning ability is largely preserved even as safety temperature modulates expert routing.

---

> > ### Comment · Reviewer_NXtt · 2025-11-26
> >
> > Thanks for your rebuttal. Table 3 of Appendix B.4 only evaluates simple benchmarks, and I think there might be a performance penalty for more complex reasoning tasks, or Pass@k performance.

---

> ### Author Response · Authors · 2025-11-27
> **Response to Reviewer NXtt**
>
> Thank you for your suggestion. We further evaluated the UpSafe℃ under different safety temperatures $\tau$ on more complex mathematical reasoning benchmarks, including **AIME2025** and **AIME2024**. We report **pass@1 / pass@3** here. The results are as follows.
>
> ***AIME2025, pass@1 / pass@3***
>
> | Model                                 | original          | $\tau$=0.25           | $\tau$=0.5            | $\tau$=0.75           | $\tau$=1              |
> |---------------------------------------|-------------------|------------------|------------------|------------------|------------------|
> | DeepSeek-R1-Distill-Qwen-7B           | 43.33 / 56.67     | 40.00 / 50.00    | 40.00 / 53.33    | 36.67 / 53.33    | 46.67 / 56.67    |
> | DeepSeek-R1-Distill-Llama-8B          | 23.33 / 40.00     | 23.33 / 46.67    | 20.00 / 40.00    | 30.00 / 36.67    | 23.33 / 36.67    |
>
> ***AIME2024, pass@1 / pass@3***
>
> | Model                    | original    | $\tau$=0.25          | $\tau$=0.5           | $\tau$=0.75          | $\tau$=1             |
> |--------------------------|-------------------|------------------|------------------|------------------|------------------|
> | DeepSeek-R1-Distill-Qwen-7B        | 53.33 / 66.67     | 53.33 / 63.33    | 50.00 / 73.33    | 56.67 / 66.67    | 46.67 / 70.00    |
> | DeepSeek-R1-Distill-Llama-8B        | 50.00 / 60.00     | 46.67 / 60.00    | 46.67 / 63.33    | 50.00 / 56.67    | 50.00 / 63.33    |
>
> Based on the results, we observe that varying the safety temperature $\tau$ does not significantly degrade performance on complex benchmarks such as AIME. In some settings, the performance even improves (as also noted by Q2 for Reviewer sSyc), which may seem unexpected. We offer the following analysis.
>
> This behavior can be explained by the design of UpSafe℃  from dense to sparse . Varying the safety temperature directly alters the stochasticity of token routing:
>
> 1. For small $\tau$, the routing tends to consistently select the general expert.
>
> 2. As $\tau$ increases, the routing distribution becomes less concentrated. Under our 1 → 4 upcycling setup (one general + three safety experts), higher $\tau$ expands the effective choice space for tokens and increases the probability of selecting among the three safety experts. This alters the stochasticity of token routing, which in turn influences the final token logits and ultimately impacts the model’s performance on AIME.
>
> This observation is also consistent with findings reported in prior work [1], which shows that for medium-sized models (e.g., DeepSeek-R1-Distill-Qwen-7B), higher sampling temperatures (i.e., stochasticity) can lead to improved AIME performance. In addition, [1] also reports that for larger-scale models (e.g., 70B), performance tends to degrade as stochasticity increases. Therefore, extending UpSafe℃ to larger models—such as 70B—would be an important direction for future work.
>
> **Reference**
>
> [1] https://github.com/GAIR-NLP/AIME-Preview

---

### Official Review · Reviewer_pjvY · 2025-10-27

**Soundness:** 3
**Presentation:** 3
**Contribution:** 2
**Rating:** 4
**Confidence:** 4

**Summary:**

This paper proposes UPSAFE°C, a safety-alignment method that upcycles a dense transformer into a sparse MoE by identifying safety-critical layers, duplicating their MLP blocks into safety experts, and introducing a learnable router to gate between general and safety experts. The training is done in two stages: first, fine-tuning the safety experts on harmful data, then training the router on a mixture of harmful and STAR-benign prompts. At inference time, a “safety temperature” controls how much the router favors safety experts versus the general expert, aiming to enable controllable safety–utility trade-offs without retraining. The approach is framed as lightweight and modular, offering adjustable refusal strength at runtime.

**Strengths:**

The proposed method is conceptually neat: instead of retraining the entire model, it inserts a sparse MoE structure into safety-critical layers and uses routing as a soft guardrail. The idea of tuning a temperature at inference for dynamic safety–utility control is intuitive and flexible. The method is orthogonal to traditional gradient-based alignment methods, potentially allowing fast safety interventions without modifying the main model. It also engages with an increasingly relevant direction—how to build modular, controllable safety mechanisms inside model architectures rather than relying solely on external filtering.

**Weaknesses:**

1. The evaluation omits several widely used and stronger baselines, including Antidote: Post-fine-tuning Safety Alignment for Large Language Models Against Harmful Fine-tuning Attack, Safe LoRA: the Silver Lining of Reducing Safety Risks when Fine-tuning Large Language Models, Shape it Up! Restoring LLM Safety during Finetuning via STAR-DSS, Safety Alignment Should Be Made More Than Just a Few Tokens Deep, and Lisa: Lazy Safety Alignment for Large Language Models against Harmful Fine-tuning Attack. More importantly, the paper does not compare against a simple guardrail + rejection sampling baseline, which remains standard and surprisingly effective in practice.
2. There is no quantitative discussion of how many parameters are added through expert duplication and router layers. This is crucial because adding multiple safety experts per layer can substantially increase model size and memory footprint, undermining claims of lightweight deployment.
3. Missing capability–safety trade-off evaluation: The safety expert training uses only harmful and STAR benign data, but no general capability data (e.g., math or reasoning tasks). This makes it impossible to assess whether MoE upcycling slows learning or harms utility compared to dense fine-tuning. STAR benign prompts are too narrow, which explains the high over-refusal rate observed — the router learns a conservative boundary that over-activates safety experts.
4. Limited applicability to MoE backbones: The method assumes a dense base model. If the base model is already MoE (e.g., Mixtral, DeepSeek-MoE), it is unclear whether to (a) add new safety experts or (b) repurpose existing ones, and either approach would require router retraining and disrupt expert balancing. The paper does not discuss this scenario, which is critical for modern production-scale models.
5. Aadaptive attacks: The router can be bypassed or manipulated through adaptive prompt strategies. For example: A malicious prompt can be crafted to appear benign at the start, routing to the general expert and bypassing safety. Adversarial fine-tuning could shift router decision boundaries. Mixed benign–harmful inputs could remain in the general expert’s path throughout the rollout.
6. Unclear RLHF training results: The method is only evaluated under SFT. In reinforcement learning settings, router activation is dynamic and not guaranteed to consistently route harmful tokens to safety experts. RL can easily shift router distributions or deactivate safety pathways over training, fundamentally breaking the controllability guarantee.
7. Scalability is unclear: Upcycling requires per-layer safety-critical analysis and expert duplication. This process must be repeated for each new model or checkpoint, unlike simple fine-tuning or guardrail filtering which scale more naturally across deployments.

**Questions:**

See above

---

> ### Author Response · Authors · 2025-11-23
> **Response to Reviewer pjvY (1/3)**
>
> > **W1: The evaluation omits several widely used and stronger baselines, ... , More importantly, the paper does not compare against a simple guardrail + rejection sampling baseline, which remains standard and surprisingly effective in practice.**
>
> Thank you for the insightful feedback regarding additional baselines.
> We would like to clarify that the methods mentioned by the reviewer—Antidote, SafeLoRA, STAR-DSS, etc.—address a **complementary** stage of the LLM lifecycle. These techniques are designed to preserve safety during potentially harmful downstream fine-tuning, whereas our work focuses on constructing a safer model through safety-oriented training. **In practice, they can be directly combined with our approach**. Below, we provide a detailed explanation.
>
> Our method focuses on **safety fine-tuning**:
>
> * We aim to train a safer model using carefully curated and safety-oriented datasets. The goal is to align the model’s internal behavior and improve its inherent safety-utility trade-off.
>
> The cited baselines focus on **safe fine-tuning**：
> * These methods aim to prevent safety degradation when a third party later fine-tunes the model with harmful or adversarial data. Their mechanisms activate during user-side adaptation.
>
> Because the two categories address different stages of the LLM lifecycle, a direct comparison would not accurately reflect their intended use cases. Instead of competing, they form two complementary lines of defense: our method strengthens intrinsic safety, while safe fine-tuning protects safety during downstream adaptation.
>
> To further clarify this relationship, we additionally provide an experiment demonstrating that downstream protection can be seamlessly combined with our approach. We adopt the simplest representative (discussed in [1])—rejection sampling—and apply it during downstream harmful fine-tuning. Importantly, in the downstream task-specific fine-tuning stage of UpSafe°C, only the original dense model needs to be updated; the safety experts and the routers are entirely frozen, as they encode fixed safety capabilities that do not participate in downstream adaptation.
>
> **Experimental setup**: For downstream fine-tuning, we use the GSM8K-PureBad mixed dataset with a 9:1 benign-to-harmful ratio from STAR-DSS Github repository [2]. We use LlamaGuard3-8B to conduct rejection sampling (RS). All experiments are conducted on Llama-3.1-8B-Instruct.
>
> |           Model            | WildJailbreak     | GSM8k            |
> |-------------------------|:----------------:|:----------------:|
> | Vanilla                    |      65.20       |      73.06       |
> | SFT                        |      77.60       |      70.26       |
> | SFT + downstream FT        |  33.60 (-44.00)  |  86.33 (+16.07)  |
> | Ours                       |      90.80       |      70.07       |
> | Ours + downstream FT       |  76.80 (-14.00)  |  90.57 (+20.50)  |
> | Ours + downstream FT + RS  |   83.00 (-7.80)  |  90.79 (+20.72)  |
>
> The results show:
>
> 1. Our model suffers substantially less safety degradation under harmful fine-tuning compared to a standard SFT model.
> 2. Adding rejection sampling (representative of the reviewer’s listed methods) provides an additional safety boost, confirming direct compatibility.
>
> Thus, we view these two classes of techniques as complementary rather than competing, and we will expand the discussion and citations of these safe fine-tuning baselines in the revised manuscript.
>
> > **W2: There is no quantitative discussion of how many parameters are added through expert duplication and router layers.**
>
> We thank the reviewer for pointing out the importance of quantifying parameter overhead.
> We would like to clarify that the parameter counts of both the duplicated safety experts and the router layers are reported in **Table. 2 of Appendix A.1**. Importantly, although naively adding safety experts to all transformer layers would indeed lead to a substantial increase in model size, our method explicitly avoids this issue. This is precisely why we introduce the safety-layer scan mechanism: it identifies only a small subset of safety-critical layers for upcycling. This design makes our approach analogous to a lightweight PEFT-style method rather than full-layer MoE expansion.
> As a result, the actual parameter increase is modest. For the 8B model, the total number of additional parameters introduced by the upcycled experts and routers is less than 20%.
> We will make this quantitative comparison more explicit in the revised manuscript to avoid confusion.
>
> **Reference:**
>
> [1] Peng, ShengYun, et al. "Shape it Up! Restoring LLM Safety during Finetuning." NeurIPS (2025).
>
> [2] https://github.com/poloclub/star-dss

---

> > ### Author Response · Authors · 2025-11-23
> > **Response to Reviewer pjvY (2/3)**
> >
> > > **W3: Safety-expert training does not include general capability data, making it unclear whether MoE upcycling harms utility or slows learning, and that STAR benign prompts may be too narrow, leading to over-refusal.**
> >
> > We appreciate the reviewer’s insightful comments. Here we clarify why our training design does not impair model utility and how our framework naturally separates capability learning from safety specialization.
> >
> > **First**, our primary objective is to align the model for safety while preserving its original general capabilities. To achieve this, we freeze the original MLP (the original expert) and construct safety experts by duplicating and initializing them from the same weights. During training, only the safety experts are updated with safety-oriented data, while the original experts remain untouched.
> >
> > This design naturally induces a division of labor:
> > * Original experts retain and handle general capabilities.
> > * Safety experts, initialized from the same pre-trained weights, inherit general competence but specialize toward safe behaviors with targeted safety data.
> > * The router learns to route safety-related inputs to the safety experts while preserving the routing to original experts for regular utility tasks.
> >
> > As shown in Figure 5, routing on general tasks continues to be dominated by the original experts, confirming that the model’s base capabilities remain anchored in the original backbone. Correspondingly, the utility metrics (MMLU, Math500, HumanEval) also demonstrate that adding safety experts does not degrade general capabilities.
> >
> > **Second**, if the model is later fine-tuned on downstream tasks, our design allows the user to simply fine-tune the dense backbone only, while freezing the safety experts and router. This ensures that downstream learning proceeds exactly as in a standard dense model, without significantly interfering with the safety alignment we introduce. As discussed in our response to W1, the learning of UpSafe℃ on GSM8K remains unaffected.
> >
> > **Finally**, we agree that increasing the diversity of benign/general prompts—especially near-boundary examples—can further refine the router and reduce over-refusal. This is a valuable direction, and our method is fully compatible with such improvements.
> >
> > > **W4: Limited applicability to MoE backbones**
> >
> > We appreciate the reviewer’s insightful comments. For MoE backbones, the key to performing safety upcycle is to appropriately select the participating experts and initialize the safe router such that the upcycled model preserves the original performance prior to safety training. Below we provide one feasible implementation. In our implementation, we maintain the general upcycling and training framework proposed in our paper.
> >
> > We first conduct the safety scan to identify safety-critical experts instead of layers through linear projection, thereby selecting `num_safety_expert` candidates. During upcycling, we copy these experts and correspondingly extend the existing router. Specifically, we copy the router and modify its final layer i.e.,
> >
> > `safety_router.weight = router.weight.copy()[:, [safety_expert_index]]`
> >
> > `router_score = cat([router(hidden), safety_router(hidden)]) + cat([router_bias, safety_router_bias])`.
> >
> > Therefore, the routing logits now cover both the original experts and the newly added safety experts, yielding num_expert + num_safety_expert routing scores.
> >
> > We set the bias terms associated with the added safety experts to −1, ensuring that they are deprioritized during initial top‑k expert selection.
> > As a result, the upcycled MoE model contains additional safety experts while remaining equivalent to the MoE backbone before safety training. Fine-tuning then proceeds identically to the dense case described in our main method.
> >
> > This extension to MoE backbone is an important part of our future work and we will discuss it in the revised manuscript.

---

> > > ### Author Response · Authors · 2025-11-23
> > > **Response to Reviewer pjvY (3/3)**
> > >
> > > > **W5: The router can be bypassed or manipulated through adaptive prompt strategies and adversarial fine-tuning could shift router decision boundaries.**
> > >
> > > We appreciate the reviewer's concern regarding adaptive manipulation of the router. We clarify that our experiments have explicitly evaluated adversarial prompt attacks. In addition to WildJailbreak (reported in the paper), we have included further results on PAIR and ReNeLLM adaptive jailbreaks in the GR1. Across all three settings, the model shows strong robustness. One key reason is that our router operates per-token, enabling fine-grained and dynamic routing rather than relying on a brittle prompt-level classifier, which empirically makes adaptive bypassing substantially more difficult.
> > >
> > > Regarding adversarial fine-tuning attacks: as discussed in our response to W1, our design inherently mitigates this risk. The safety experts and router are added through MoE upcycling and are trained in a PEFT-like manner. During downstream fine-tuning for new tasks, only the original dense model parameters are updated, while the router and safety experts remain frozen. Therefore, adversarial or task-specific fine-tuning does not shift the router’s decision boundaries and cannot significantly compromise its safety behavior.
> > >
> > > We will clarify these points and include the additional adaptive attack results in the revised manuscript.
> > >
> > > > **W6: Unclear RLHF training results**
> > >
> > > Thank you for raising this point. Ensuring stable routing under RLHF is indeed a known challenge for MoE-based architectures, and recent works have also highlighted issues such as training–inference mismatch and routing stability issues in MoE under RL [1, 2]. This is an active research direction.
> > >
> > > We believe that upcycling can be naturally extended to RLHF frameworks. One straightforward idea is to adopt algorithms like GSPO and add light regularization on the router to stabilize safety-expert activation during RL optimization. Systematically exploring RLHF for upcycled models is a promising and technically interesting direction, and it is part of our planned future work.
> > >
> > > > **W7: Scalability is unclear: Upcycling requires per-layer safety-critical analysis and expert duplication.**
> > >
> > > We note that the main additional cost of upcycling comes from the safety-critical layer analysis. However, this scan is lightweight—typically around 15 minutes for a new model—and only needs to be performed once per model family. For widely used models, such results can be published and reused by the community. The parameter increase introduced by upcycling is also modest (e.g., <20% for 8B models), making the method practical and scalable in deployment.
> > >
> > > **Reference:**
> > >
> > > [1] Zheng, Chujie, et al. "Group sequence policy optimization." arXiv preprint arXiv:2507.18071 (2025).
> > >
> > > [2] Ma, Wenhan, et al. "Stabilizing MoE Reinforcement Learning by Aligning Training and Inference Routers." arXiv preprint arXiv:2510.11370 (2025).

---

> > > > ### Comment · Reviewer_pjvY · 2025-11-26
> > > >
> > > > Thanks for your response. I believe a few points are still missing in the response.
> > > >
> > > > 1. There is still no clear parameter breakdown, no FLOPs analysis, and no comparison to LoRA / SafeLoRA-style PEFT methods. Without explicit numerical accounting of expert duplication and router expansion, it is not possible to evaluate whether UPSAFE°C is truly lightweight, especially relative to other PEFT-based safety methods. Thus, this core concern about overhead and scalability remains insufficiently answered.
> > > >
> > > > 2. Although new results claim PAIR and ReNeLLM results were added, it is unclear how these evaluations were conducted, what settings were used, or whether the router is actually robust to early-token or prefix-based manipulations. Moreover, the rebuttal’s argument about freezing the router still does not resolve the fundamental issue that downstream fine-tuning can shift the hidden representations and therefore alter routing behavior, even if router weights are fixed. As a result, the vulnerability of the router to bypass or drift under adversarial pressure remains only partially addressed.
> > > >
> > > > 3. The rebuttal reframes the concern (Guardrail + Rejection Sampling) as “different lifecycle stages” instead of addressing the actual request: evaluating UPSAFE°C against standard guardrail + rejection sampling **at inference**, which is the reviewer’s main point. The provided downstream fine-tuning experiment does not substitute for a direct comparison on the same safety benchmarks. As a result, it remains unclear whether UPSAFE°C meaningfully outperforms the simplest and most widely used baseline.

---

> > > > > ### Author Response · Authors · 2025-12-01
> > > > > **Response to Reviewer pjvY**
> > > > >
> > > > > > **Q1: There is still no clear parameter breakdown, no FLOPs analysis, and no comparison to LoRA / SafeLoRA-style PEFT methods.**
> > > > >
> > > > > We thank the reviewer for the question. We would like to first clarify that our motivation is not to compare UpSafe℃ with PEFT-style methods in terms of being “lightweight.” Our goal is to achieve a highly balanced safety–utility tradeoff while enabling dynamic safety during inference. With this broader motivation, Safety Layer Scan effectively reduces the parameter overhead by selecting only safety-critical layers for upcycling, making our approach substantially lighter than the per-layer upcycle design noted by the reviewer in Q3. We only adopt a PEFT-like implementation, rather than aiming to compete with PEFT methods on lightweightness.
> > > > >
> > > > > Below, we provide the FLOPs and parameter breakdown of UpSafe℃ on Llama-3.1-8B-Instruct:
> > > > >
> > > > > | Model    | Total Params (B) | Activated Params (B) | MACs (T) | FLOPs (T) |
> > > > > | -------- | ---------------- | ------------------------------- | -------- | --------- |
> > > > > | Original | 8.03             | 8.03                            | 0.965    | 1.93      |
> > > > > | UpSafe℃  | 9.62             | 8.56                           | 1.03     | 2.07      |
> > > > >
> > > > >
> > > > > | Component                | Value            |
> > > > > |--------------------------|------------------|
> > > > > | Safety Layers            | 13, 17, 18       |
> > > > > | Safety Experts per Layer | 176.16M × 3      |
> > > > > | Router                   | 16.38K           |
> > > > > | Total per Layer          | 528.48M          |
> > > > > | Total                    | 1585.50M (1.59B)        |
> > > > >
> > > > > From the results, we observe that the overhead introduced on the 8B model is modest. The total parameter increases by 19.8%, and the number of activated parameters during inference increases by only 6.6%, while the FLOPs rise slightly (1.93T → 2.07T). This shows that UpSafe℃ strengthens safety while maintaining efficient inference.
> > > > >
> > > > > >**Q2: Although new results claim PAIR and ReNeLLM results were added, it is unclear how these evaluations were conducted, what settings were used, or whether the router is actually robust to early-token or prefix-based manipulations. Moreover, the rebuttal’s argument about freezing the router still does not resolve the fundamental issue that downstream fine-tuning can shift the hidden representations and therefore alter routing behavior, even if router weights are fixed.**
> > > > >
> > > > > We thank the reviewer for the question.
> > > > >
> > > > > **First**, we implemented PAIR and ReNeLLM using the AISafetyLab [1] framework and tested them on 100 samples from the AdvBench dataset..
> > > > >
> > > > > **Second**, to address the reviewer’s concern regarding prefix-based router manipulation, we conducted additional experiments on Llama-3.1-8B-Instruct using prefill attacks [2] of varying prefix lengths (0 → 5 → 10 → 25 → 50 tokens).
> > > > > The measured harmful rates (%) are:
> > > > > | Methods        | 0  | 5  | 10 | 25 | 50 |
> > > > > | ------------ | -- | -- | -- | -- | -- |
> > > > > | **Original** | 20 | 34 | 42 | 48 | 58 |
> > > > > | **SFT**      | 0  | 6  | 16  | 26 | 34 |
> > > > > | **Ours**     | 0  | 12  | 14  | 18  | 26 |
> > > > >
> > > > > These results show that the router retains strong robustness under prefix attack.
> > > > >
> > > > > **Third**, regarding the reviewer’s concern that downstream fine-tuning may shift the hidden representations—even if the router is frozen—we offer two clarifications:
> > > > >
> > > > > 1. **Benign downstream fine-tuning**. For benign task (e.g., math, code), the changes in hidden representations tend to remain within the model’s safe representational manifold. Because our router is trained as a pre-defined safety boundary (and then frozen), these small, task-aligned shifts do not meaningfully alter the routing behavior. As a result, the model’s safety performance remains stable under benign fine-tuning.
> > > > >
> > > > > 2. **Harmful or adversarial fine-tuning**. We agree that this falls into the broader category of safety drift. As discussed in our response to W1, UpSafe℃ can be naturally combined with safe finetuning approaches mentioned by the reviewer. Our experiments with Rejection Sampling already demonstrate that this combination is effective in preserving routing behavior under harmful fine-tuning pressure.
> > > > >
> > > > > Overall, this issue lies beyond the scope of UpSafe℃ itself and relates to the broader problem of lifecycle safety governance. We view UpSafe℃ as a foundational component that can be integrated with future drift-mitigation techniques, and we consider this a valuable direction for further exploration.
> > > > >
> > > > > **Reference**
> > > > >
> > > > > [1] https://github.com/thu-coai/AISafetyLab
> > > > >
> > > > > [2] https://github.com/haizelabs/llama3-jailbreak

---

> > > > > > ### Author Response · Authors · 2025-12-01
> > > > > > **Response to Reviewer pjvY**
> > > > > >
> > > > > > >**Q3: Evaluating UpSafe℃ against standard guardrail + rejection sampling at inference**
> > > > > >
> > > > > > We thank the reviewer for the clarification. **As noted in our response to Reviewer TAMx (W1)**, we have already conducted this experiment. Below we restate the results for clarity.
> > > > > >
> > > > > > We employ LlamaGuard3-8B to filter inputs on three harmful benchmarks. Inputs judged as unsafe are directly rejected, and only the remaining inputs are passed to the model (Llama-3.1-8b-Instruct) for safety evaluation. The results are shown below:
> > > > > >
> > > > > > |      Methods       |   JBB   | StrongREJECT | WildJailbreak |
> > > > > > |:----------------:|:-------:|:------------:|:--------------:|
> > > > > > | Filtering-based  |  100    |     100      |     79.60      |
> > > > > > | Ours (0.5)       |  100    |     100      |     90.80      |
> > > > > >
> > > > > > Our method achieves stronger robustness on WildJailbreak. Moreover, unlike LlamaGuard3—which provides an **external**, pre-processing–only safety filter—our approach enhances the model’s **intrinsic** safety.

---

### Author Response · Authors · 2025-11-23
**General Response**

We thank all reviewers for their efforts and constructive comments. Regarding the reviewers' common concerns, we provide the following general responses.
> **GR1: Additional baselines and broader jailbreak evaluations**

We have incorporated more baseline comparisons and expanded the set of jailbreak attacks in our evaluation. For LLM settings, we added **Circuit Breakers (CB) [1]** and **Representation Bending (RepBend) [2]** as baselines; for LRM settings, we included **SafePath [3]** and **SafeKey [4]**. We also introduced two additional adaptive jailbreak attacks, including **PAIR [5]** and **ReNeLLM [6]**. The added results consistently support the robustness and effectiveness of our method.

**Experimental setup**: For LLM evaluations, we use Llama-3-8B-Instruct; for LRM evaluations, we use DeepSeek-R1-Distill-Llama-8B. We report UpSafe°C performance at $\tau = 0.5$ and $\tau = 0.8$.
___
**Llama-3-8B-Instruct**
___
|      Methods      | StrongREJECT |   JBB   | WildChat | WildJailbreak |  PAIR  | ReNeLLM | Avg.Safety | |  XStest  | |  MMLU  | Math500 | HumanEval | Avg.General |
|:---------------:|:------------:|:-------:|:--------:|:--------------:|:------:|:-------:|:-----------:|:--:|:--------:|:--:|:------:|:-------:|:---------:|:------------:|
|       CB        |   100.00     | 100.00  |  82.70   |     89.00      | **97.00**  |  74.00  |    90.45    | |  95.20   | | 61.01  |  23.20  |   51.80   |    45.34    |
|     RepBend     |    99.36     |  99.00  |  86.76   |     88.20      | 96.00  |  62.50  |    88.64    | |  93.20   | | 63.05  |  16.80  |   53.70   |    44.52    |
|    Ours(0.5)    |   100.00     | 100.00  |  89.46   |     90.80      | 94.00  |  76.00  |    91.71    | |  **95.20**   | | **63.95**  |  **27.40**  |   **57.30**   |    **49.55**    |
|    Ours(0.8)    |   **100.00**     | **100.00**  |  **91.08**   |     **93.20**      | 96.00  |  **80.00**  |    **93.38**    | |  93.20   | | 62.87  |  27.20  |   55.60   |    48.56    |
___
**DeepSeek-R1-Distill-Llama-8B**
___
|      Methods      | StrongREJECT |   JBB   | WildChat | WildJailbreak |  PAIR  | ReNeLLM | Avg.Safety | |  XStest  | |  MMLU  | Math500 | HumanEval | Avg.General |
|:---------------:|:------------:|:-------:|:--------:|:--------------:|:------:|:-------:|:-----------:|:--:|:--------:|:--:|:------:|:-------:|:---------:|:------------:|
|    SafeKey      |   100.00     | 100.00  |  72.70   |     82.00      | 70.00  |  59.00  |    80.95    | |  78.20   | | 71.58  | 67.40  |   70.39   |    69.79    |
|    SafePath     |    99.04     | 100.00  |  80.76   |     72.40      | 74.00  |  83.00  |    84.20    | |  64.00   | | 70.43  | 70.40  |   68.60   |    69.81    |
|   Ours(0.5)     |   100.00     | 100.00  |  75.68   |     70.40      | 69.00  |  78.00  |    82.18    | |  **82.80**   | | 72.00  | **78.40**  |   **76.70**   |    **75.70**    |
|   Ours(0.8)     |   **100.00**     | **100.00**  |  **82.74**   |     **86.40**      | **77.00**  |  **83.00**  |    **88.19**    | |  79.20   | | **72.20**  | 78.20  |   76.30   |    75.57    |

Across the expanded set of baselines and attacks, UpSafe°C  provides a more flexible and practical balance between safety and utility compared to one-size-fits-all baseline defenses.

**Reference**:

[1] Zou, Andy, et al. "Improving alignment and robustness with circuit breakers." NeurIPS (2024).

[2] Yousefpour, Ashkan, et al. "Representation bending for large language model safety." ACL (2025).

[3] Jeung, Wonje, et al. "SAFEPATH: Preventing Harmful Reasoning in Chain-of-Thought via Early Alignment." NeurIPS (2025).

[4] Zhou, Kaiwen, et al. "Safekey: Amplifying aha-moment insights for safety reasoning." EMNLP (2025).

[5] Chao, Patrick, et al. "Jailbreaking black box large language models in twenty queries." 2025 IEEE Conference on Secure and Trustworthy Machine Learning (SaTML). IEEE, 2025.

[6] Ding, Peng, et al. "A wolf in sheep's clothing: Generalized nested jailbreak prompts can fool large language models easily." NAACL (2023).

---

### Author Response · Authors · 2025-12-02
**Summary of Review, Rebuttal, and Discussion**

Dear Area Chair and Reviewers,

As the active discussion phase had to be closed due to the privacy-leak, we summarize the key points from the reviews and our rebuttal to support a fair, transparent, and comprehensive assessment of our work.

We thank all reviewers for actively participating in the discussion and providing valuable feedback that significantly strengthened the paper. **Reviewer sSyc raised the score from 6 to 8** (23 Nov 2025, before the leak incident) after considering our expanded baselines and clarifications. **Reviewer TAMx also increased the score from 4 to 6** (24 Nov 2025, before the leak incident)  following our additional experiments and detailed responses. **We also engaged in multiple rounds of discussion with Reviewer pjvY and Reviewer NXtt**; although the discussion phase closed before they could submit further replies, we believe our rebuttal directly addressed the most concerns they raised. We believe the discussion has substantially improved the clarity, completeness, and positioning of our contributions.

**We have revised our paper according to the discussion (highlighted in blue text color). The modifications are summarized as follows.**

1. (For Reviewer sSyc, TAMx and  pjvY) We have incorporated more baseline comparisons and expanded the set of jailbreak attacks in our evaluation in **Appendix. B.5**.

2. (For Reviewer TAMx and  pjvY) We have provided a more detailed analysis of the computational and parameter overhead introduced by our method, including comparisons of parameter counts, MACs/FLOPs, and training/inference efficiency in **Appendix. B.7**.

3. (For Reviewer NXtt and  pjvY) We further present additional experiments demonstrating that our method preserves general utility performance, and we analyze the effects of different safety temperatures on more challenging reasoning tasks in **Appendix. B.4**.

4. (For Reviewer pjvY) We have added an extended discussion comparing our approach with safe finetuning–based methods, clarifying conceptual differences, complementary strengths, and practical implications in **Appendix. B.6**.

5. (For Reviewer sSyc and NXtt) We have refined the description of our safety-critical layer scan method to make the motivation and contribution clearer in **Section. 2.1**.

6. (For Reviewer sSyc and pjvY) We have refined the analysis and presentation of our safety temperature experiments, providing clearer explanations of its behavior and its impact on the safety–utility trade-off in **Section. 3.2**.

We have also addressed several reviewer concerns and misunderstandings directly in the rebuttal; please refer to the one-on-one discussions below for full details.

**In addition to the concerns addressed above, we sincerely appreciate that the reviewers also highlighted several key strengths of our work**:

1. Overall of our method.
Reviewers described our approach as “conceptually neat” (pjvY) and “a creative and underexplored idea” (NXtt), acknowledging the novelty of our safety-aware upcycling framework for controllable alignment.

2. Safety-Critical Layer Scan.
Reviewers noted that our proposed layer selection mechanism offers "clear insights into how and where safety improvements occur" (sSyc) and “efficiently identifies layers most responsive to safety signals—avoiding unnecessary parameter tuning across all layers” (TAMx). We appreciate this recognition, as the scan procedure is central to keeping the method efficient and targeted.

3. Safety-Aware Upcycling.
Reviewers emphasized that the upcycling mechanism “enables fast safety interventions without modifying the main model” (pjvY) and  "specializes safety experts in mitigating harmful content while enabling the router to discriminate between benign and harmful inputs, striking a robust balance between safety enhancement and general capability preservation" (TAMx). This captures the motivation behind designing a safety path that can be added modularly while preserving core capabilities.

4. Safety Temperature.
Reviewers also commended the safety temperature mechanism, describing it as “intuitive and flexible” (pjvY) and highlighting that it “provides fine-grained, dynamic control over the safety–utility trade-off at inference time” (TAMx). This positive feedback affirms our goal of enabling controllable and adjustable safety behaviors.

5. Experimental Results.
Reviewers recognized that our experimental results “show strong performance across both standard LLMs and LRMs” (sSyc), and that our method “demonstrates strong generalization across model types, scales, and challenging benchmarks, outperforming baselines in both safety improvements” (TAMx).

Thank you again for your time, effort, and service to the community.

Best regards,

Authors

---

### Meta-Review · Area_Chair_nDTi · 2025-12-06

**Summary:**

The main concerns that drive a reject leaning are the strength and completeness of the empirical validation, the unclear practical overhead and scalability of the method, and open questions about novelty and robustness. Several reviewers agree that, although the idea of safety aware upcycling with a safety temperature is conceptually appealing, the evaluation does not convincingly show that it is better than strong and simple baselines such as guardrail plus rejection sampling or modern safe finetuning methods.

There are also worries about how the approach scales to already MoE based backbones and to RLHF style training, how robust it is under adaptive attacks that can bypass the router, and whether the very strong reported numbers are fully reliable. Overall, the paper sits in the marginal zone: interesting direction and promising experiments, but key questions remain unresolved enough that the work does not clearly cross the bar for acceptance. Therefore, my decision for this paper is Reject.

**Reviewer Concerns:**

In the rebuttal period, several concerns appear to be substantially addressed and the revised draft. The authors expanded the set of baselines and attacks, adding comparisons to more jailbreak methods and to additional safety finetuning approaches, which directly responds to points from reviewers sSyc, TAMx, and pjvY. They also provided a more detailed breakdown of parameter counts, compute and memory overhead, and training and inference efficiency, which targets the practicality concerns from TAMx and pjvY. Additional experiments on general utility benchmarks and more difficult reasoning tasks, together with a refined analysis of how the safety temperature affects performance, speak to the safety utility tradeoff worries raised by NXtt and sSyc. The description of the safety critical layer scan and the role of the safety temperature were clarified and better motivated in the main text and appendices, which helps with understanding and presentation issues.
But Several key issues remain unclear. The paper still lacks comparison to a simple and widely used guardrail plus rejection sampling baseline, despite explicit requests. Scalability to MoE backbones is discussed only conceptually, with no evidence showing how added safety experts interact with routing or load balancing. The stability of the router under RLHF remains untested, leaving open whether the safety pathway would hold during further training. Adaptive attacks that can bypass the router are not systematically evaluated, so robustness is still uncertain. Questions about novelty beyond structural adaptation also persist, and the unusually high safety scores are not fully explained or validated. These gaps justify maintaining a cautious stance.

**Reviewer Scores:**

Reviewer pjvY had strong reservations about missing baselines, adaptive attacks, MoE backbones, and RLHF; the rebuttal addresses some of these partially but not decisively, the reviewer responded back, but not fully engaged. I would expect 4->4

Reviewer NXtt worried about novelty and the impact of the safety temperature on complex reasoning. The reviewer participate in the discussion.

Reviewer TAMx  likely keep the updated score.

Reviewer sSyc already raised their score to 8 after the rebuttal, 8->8.

---

### Decision · Program_Chairs · 2026-01-26

Reject